# PROCEEDINGS A

computational biology, computational mathematics

COVID-19, modelling, facemask

**Author for correspondence:**
Richard O. J. H. Stutt
e-mail: rs481@cam.ac.uk

# A modelling framework to assess the likely effectiveness of facemasks in combination with 'lock-down' in managing the COVID-19 pandemic

Richard O. J. H. Stutt[1], Renata Retkute[1], Michael Bradley[2], Christopher A. Gilligan[1] and John Colvin[3]

[1]Epidemiology and Modelling Group, Department of Plant Sciences, University of Cambridge, Downing Street, Cambridge CB2 3EA, UK
[2]The Wolfson Centre for Bulk Solids Handling Technology, and
[3]Natural Resources Institute, University of Greenwich, Chatham Maritime ME4 4TB, UK

ROJHS, 0000-0002-1765-2633

COVID-19 is characterized by an infectious pre-symptomatic period, when newly infected individuals can unwittingly infect others. We are interested in what benefits facemasks could offer as a non-pharmaceutical intervention, especially in the settings where high-technology interventions, such as contact tracing using mobile apps or rapid case detection via molecular tests, are not sustainable. Here, we report the results of two mathematical models and show that facemask use by the public could make a major contribution to reducing the impact of the COVID-19 pandemic. Our intention is to provide a simple modelling framework to examine the dynamics of COVID-19 epidemics when facemasks are worn by the public, with or without imposed 'lock-down' periods. Our results are illustrated for a number of plausible values for parameter ranges describing epidemiological processes and mechanistic properties of facemasks, in the absence of current measurements for these values. We show that, when facemasks are used by the public all the time (not just from when symptoms first appear), the effective reproduction number, $R_e$, can be decreased below 1, leading to the mitigation of epidemic spread.

Under certain conditions, when lock-down periods are implemented in combination with 100% facemask use, there is vastly less disease spread, secondary and tertiary waves are flattened and the epidemic is brought under control. The effect occurs even when it is assumed that facemasks are only 50% effective at capturing exhaled virus inoculum with an equal or lower efficiency on inhalation. Facemask use by the public has been suggested to be ineffective because wearers may touch their faces more often, thus increasing the probability of contracting COVID-19. For completeness, our models show that facemask adoption provides population-level benefits, even in circumstances where wearers are placed at increased risk. At the time of writing, facemask use by the public has not been recommended in many countries, but a recommendation for wearing face-coverings has just been announced for Scotland. Even if facemask use began after the start of the first lock-down period, our results show that benefits could still accrue by reducing the risk of the occurrence of further COVID-19 waves. We examine the effects of different rates of facemask adoption without lock-down periods and show that, even at lower levels of adoption, benefits accrue to the facemask wearers. These analyses may explain why some countries, where adoption of facemask use by the public is around 100%, have experienced significantly lower rates of COVID-19 spread and associated deaths. We conclude that facemask use by the public, when used in combination with physical distancing or periods of lock-down, may provide an acceptable way of managing the COVID-19 pandemic and re-opening economic activity. These results are relevant to the developed as well as the developing world, where large numbers of people are resource poor, but fabrication of home-made, effective facemasks is possible. A key message from our analyses to aid the widespread adoption of facemasks would be: 'my mask protects you, your mask protects me'.

## 1. Introduction

The current COVID-19 pandemic, caused by the virus species *severe acute respiratory syndrome-related coronavirus*, named SARS-CoV-2 [1], has stimulated considerable controversy over the potential benefits of facemask use by the public and the timing of the initiation and termination of 'lock-down' periods. We define 'facemask' here to mean a protective covering for the nose and mouth, designed to interfere with airborne pathogen transmission.

Clear answers to the questions surrounding facemask use are required urgently because they could inform national governments' decisions and so prevent substantial loss of life, which might be avoided with this very 'low'-level/inexpensive technology, and minimize the risk of health systems being overwhelmed with consequent high mortality rates of medical practitioners, front-line essential workers and those involved in healthcare sectors around the globe. It is also possible that this low-level technology, including home-made masks, could reduce the severe global economic impact of COVID-19, which has the potential to cause billions of people to suffer shortened life expectancy because of a reduced standard of living [2].

SARS-CoV-2 is similar to other respiratory pathogens in that airborne transmission occurs by inhaling droplets loaded with SARS-CoV-2 particles that are expelled by infectious people who are talking/coughing/sneezing [3,4]. This is most likely to occur in poorly ventilated areas where droplets or mist particles can accumulate and be inhaled [5–7]. Infection can also occur through the mucous membranes of the head (eyes, nose and mouth), when SARS-Cov-2 particles are picked up on the hands and then transferred to the head by face-touching behaviours. The currently available control measures to combat SARS-Cov-2, therefore, include: physical distancing, population lock-down periods, good sanitation/hand washing/surface disinfecting, good ventilation, facemask and visor protection, as well as diagnostics followed by contact tracing and quarantine of infected and exposed individuals.

There is a wide range of dynamic-simulation, individual-based and statistical models that are being used to analyse COVID-19 data. Many of these models, however, are complex, so encounter challenges of analysis and interpretation in all but the most expert hands [8–15]. Our

intention here is to provide a simple modelling framework to examine the probable effectiveness of facemask wearing in combination with lock-down periods on the dynamics of COVID-19 epidemics. This involves scaling from individual behaviour to the level of populations to enable conclusions to be drawn about the effectiveness, or otherwise, of wearing facemasks to reduce the spread of SARS-CoV-2.

There is an extensive literature describing the dynamics of exhalation of virus from infected individuals (e.g. velocity, reach, separation into small and large droplets) [16,17] with analyses and models that focus on the individual. Here, we approach the key questions of this article by linking the effects of facemask wearing on the individual processes of SARS-Cov-2 transmission with population-level models to assess the effectiveness, or otherwise, of facemask adoption in combination with lock-down periods under different scenarios. We provide a framework for objective recommendations that could reduce the risks of future waves of the COVID-19 pandemic. In addition, we provide a synthesis of current knowledge and identify key areas of missing information/data that would be required to refine parameter values affected by facemask interference in SARS-CoV-2 transmission processes.

We first use an agent-based branching process model [18] to ask the simple question: given the high infectiousness of SARS-CoV-2, what level of facemask adoption by the public, associated with what level of facemask efficacy, would be required to reduce the effective reproduction number ($R_e$) to below 1? In our second model, we adapt a conventional susceptible–infected–removed (SIR) compartmental model [19,20] with the extension of including 'free-living' inoculum. The latter is shed from both pre- and post-symptomatic infectious individuals by talking, coughing and sneezing to generate virus-bearing droplets and a reservoir of SARS-CoV-2 particles on surfaces, known as fomites.

At the start of the pandemic, the World Health Organization (WHO) did not advocate facemask wearing by the public owing to concerns over efficacy and the shortage of masks and other personal protective equipment (PPE) for health workers. We argue that the lack of experimental population-based data on facemask use [21] cannot be equated with facemask ineffectiveness, particularly when it is accepted that patients with other respiratory diseases such as influenza have been recommended to wear facemasks to limit virus-particle-laden droplet spread. Influenza has different dynamics from SARS-CoV-2, including a lower effective value for transmissibility [22], yet Yan *et al*. [22], for example, also reported that an 80% compliance rate for respiratory protective devices eliminated an influenza outbreak. A modelling study associated with the influenza management strategy [22] was also useful to public-health officials making decisions concerning resource allocation or public-education strategies. An earlier study [23] also concluded that household transmission of influenza could be reduced by the use of facemasks and intensified hand hygiene, when implemented early and used diligently. In addition, concerns about the acceptability and tolerability of the interventions should not be used as a reason against their recommendation.

Our models can be used to consider the transmission dynamics from the perspective of the susceptible individual, including the possibility that facemask wearing may increase the risk of transmission (e.g. through adjustments to facemasks that result in increased touching of the face, with associated risk of inadvertent transmission of SARS-CoV-2) from airborne and fomite reservoirs. We also examine changes in parameter values across plausible ranges and seek to identify the likely range of effectiveness of facemask wearing, in association with population lock-down. In the absence of robust and reliable estimates for critical parameters, this approach can be used to provide simulations over a wide range of values and also to highlight where improved epidemiological parameter estimates are required. In this spirit, we propose, therefore, that the work presented here provides an objective and logical approach to examining the key question of whether, or not, the public should be advised to wear facemasks in the current COVID-19 pandemic. Our results hold for low-, middle- and high-income countries.

We can ask the following questions at a population level. (i) How effective and how frequently would facemasks need to be used by the public to 'flatten the disease progress curve'? (ii) How effective are facemasks at reducing the free-living SARS-CoV-2 inoculum and transmission rates?

(iii) How does the timing of the implementation of lock-down periods and facemask adoption by the public influence the models' outcomes?

## 2. Methods

We use two complementary modelling approaches to test the effectiveness of facemask wearing by sections of the population in reducing the transmission rate of SARS-Cov-2 and hence in reducing the effective reproduction number, $R_e$ (the expected number of new cases caused by a single infectious individual at a given point in the epidemic). The first model uses a branching process to investigate the reduction in transmission by wearing facemasks, in order to assess the likely effectiveness of two control variables in reducing $R_e$ for the pathogen. The control variables are the proportion of the population wearing facemasks (essentially the probability that an individual wears a mask on a given day) and the effectiveness of the mask in reducing transmission (which relates to a range of masks that extend from crude porous coverings [24,25] to masks of clinical standard [26,27]). The purpose of this model is to identify whether or not there are clear parameter ranges in which the two control variables could reduce $R_e$ sufficiently to be expected to slow or stop the epidemic spread. We simulate the consequences of wearers using facemasks routinely, or only after the onset of symptoms.

In the second model, we adapt the common SIR formulation, to which we add free-living SARS-CoV-2 particles transmitted by inhalation from droplet inoculum and by touch and contact with facial orifices from the fomite inoculum deposited on surfaces (§2b(i)). The model is used to consider the likely impacts of facemask wearing in combination with phases of lock-down, interspersed with release from them. The flexible structure of this modelling framework importantly allows a distinction to be made between the potential for facemasks to reduce transmission from infected individuals (before and after symptom expression) and the protection conferred by facemasks on susceptible individuals [28]. The latter may be positive, whereby the facemask reduces inhalation of inoculum. It may also be negative; for example, where frequent manual adjustment of the facemask increases the probability of transmission. Our intention is to provide a flexible, yet comparatively simple modelling framework to test hypotheses about facemask wearing in combination with other epidemic strategies, which also allows scaling from individual behaviour to population consequences. SARS-CoV-2 is a new disease to humanity so, given the gaps in our knowledge about certain parameter values, the inferences should be viewed in this light. Further details of the models are summarized below and the code is available at https://github.com/camepidem/COVID-19_PRSA.

### (a) Branching process transmission model

We model SARS-CoV-2 transmission as a branching process model, where the number of secondary cases caused by one infectious individual is drawn from a negative binomial distribution with the mean equal to the product of the reproduction number ($R_0$, the number of individuals infected by the introduction of a single infectious individual to an otherwise susceptible population) and dispersion parameter $k$, and the time of each new infection dependent on the incubation period of the primary case and the relative infectiousness $\beta(t)$ [18]. The random branching process has been used for many COVID-19 analyses [29–37]. Given the wide variation in quoted values for COVID-19's $R_0$ [38], we consider two cases: $R_0 = 2.2$ and $k = 0.54$ [36]; and $R_0 = 4.0$, which is in line with the recent estimate of around 3.87, which is based on the initial growth of observed cases and deaths in 11 European countries [30].

We assume that time-varying relative infectiousness follows a shifted gamma distribution, reaching a peak 1–2 days before onset of symptoms [39] and then decreasing monotonically [40,41] (figure 1). The incubation period is assumed to be lognormal with meanlog 1.43 and s.d.log 0.66 [39].

To implement control by wearing facemasks, we assumed that a proportion of the population ($p$) is wearing a mask. On a day when an individual wears a facemask, the relative infectiousness

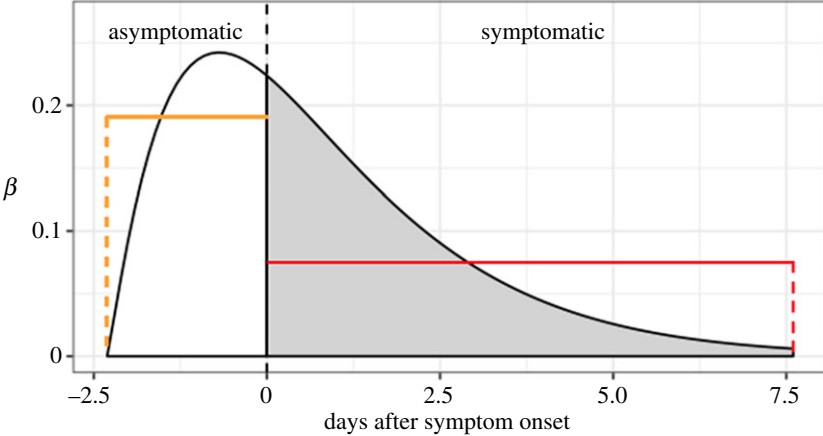

**Figure 1.** Distribution of asymptomatic and symptomatic infectiousness of COVID-19-infected individuals, used in the branching process [39]. Horizontal lines show the average infectiousness per time unit for the asymptomatic stage (orange) and the symptomatic stage (red).

is reduced by $(1 - \gamma)$, where $\gamma$ is the effectiveness of a facemask in reducing transmissibility. We explore two scenarios: wearing a facemask after the onset of symptoms and wearing a facemask all the time.

## (b) Susceptible–infected–removed model with free-living inoculum

### (i) Model description and formulation

The model structure is summarized in figure 2. There are two populations, facemask wearers and non-facemask wearers, each comprising individuals in the following categories: susceptible ($S$); exposed, i.e. latently infected ($E$); asymptomatically infectious ($I_A$); symptomatically infectious ($I_S$); and removed ($R$). The removed class includes individuals who recovered from infection and those who died. Susceptible individuals can become infected by coming into contact with inoculum produced by individuals infected with SARS-CoV-2. We separate inoculum creation by infectious individuals, which gives rise to free-living inoculum, from inoculum uptake and infection of susceptible individuals. The inoculum can either be acquired by inhaling from transient droplet ($D$) forms in the air [42] or by contact with a decaying reservoir of inoculum deposited by infected individuals in the environment as fomites ($F$) [42], which can survive for up to 72 h on some surfaces [43]. There is a rapid deposition of droplet inoculum [44–46] with a slower decay of fomite (figure 2). There are two pairs of transmission rates, therefore: $\beta_A$ and $\beta_S$ for creation of inoculum by asymptomatic and symptomatic individuals, respectively, and $\beta_D$ and $\beta_F$ for uptake and infection of susceptible individuals from the droplet and fomite inoculum, respectively. Facemask wearing affects some or all of these parameters (cf. $m_i$ in figure 2).

Facemasks reduce the amount of droplet inoculum escaping from infectious individuals [25] by capturing a proportion of droplets within the mask ($m_A$, $m_s < 1$). Facemasks also reduce the amount of droplet inoculum inhaled by susceptible individuals by capturing a proportion of droplets in inhaled air and hence reducing the uptake transmission rate ($\beta_D$) by $m_D$ (figure 2). We assume initially that masks have a negligible effect on the risk of contacting inoculum from surfaces ($\beta_F$) with $m_F = 1$. The model does, however, allow for the fact that wearing a mask could increase infection risk from fomite infection ($m_F > 1$); for example, through more frequent touching of the face when adjusting the mask. We note that significant PPE, such as a full face-hood, could act to reduce the risk of fomite infection ($m_F < 1$). In addition, sanitation interventions such as hand washing would act to reduce the risk of infection from fomites (reducing $\beta_F$) and

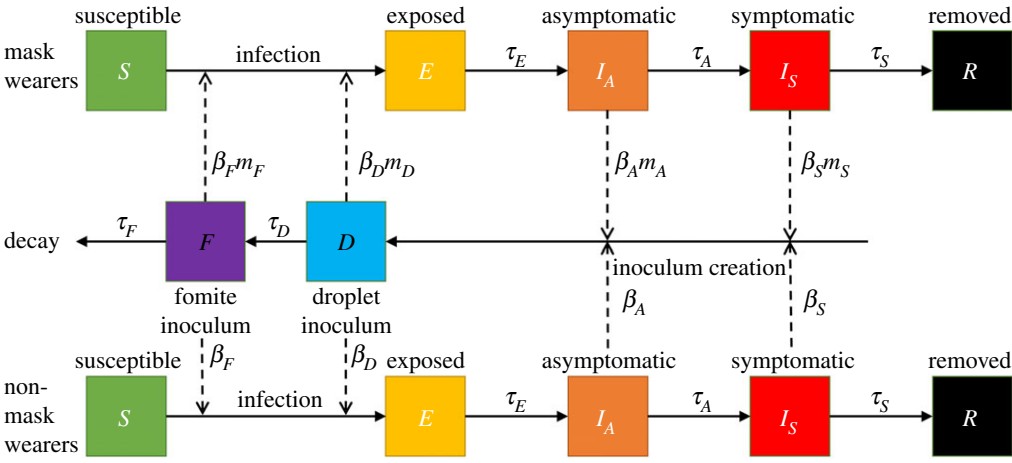

**Figure 2.** Schematic of the SIR framework model for interacting populations of facemask and non-facemask wearers, in which transmission pathways are distinguished between (i) inoculum creation by asymptomatic and symptomatic infected individuals and (ii) uptake/infection from droplet and fomite inoculum. The model can be adapted further to allow for cycles of population lock-downs.

additional cleaning of surfaces or the use of quicker self-sterilizing surfaces can be modelled via reducing the lifespan of fomite inoculum ($\tau_F$). Here, we restrict our analysis to the effects of facemasks and lock-down periods.

The model is formulated and solved as a simple deterministic differential equation model, summarized below for completeness. We note that the model can be recast readily in stochastic form with transition probabilities. It is also simple to divide the target country into metapopulations in which the contact rates differ, for example among cities and rural areas or between age classes in the population, and to spatially partition the population with localized inoculum pools. Here, we use the model to look at orders of magnitude for how facemask wearing complements a major control strategy that involves lock-down of a proportion of the population. We simulate this by assuming that lock-down reduces the transmission rates ($\beta_i$, $i = A,S,D,F$) by a fixed proportion, $q$. It reduces the inoculum produced by infectious individuals in public areas and thus reduces the inoculum available in the $D$ and $F$ pools, and additionally reduces the time spent in contact with that inoculum by susceptibles. Thus, in the model, lock-down works to reduce overall effective spread rates by a factor of $q^2$.

*Model equations*

$$\frac{dS}{dt} = -(\beta_F m_F F + \beta_D m_D D)S,$$

$$\frac{dE}{dt} = (\beta_F m_F F + \beta_D m_D D)S - \frac{E}{\tau_E},$$

$$\frac{dI_A}{dt} = \frac{E}{\tau_E} - \frac{I_A}{\tau_A},$$

$$\frac{dI_S}{dt} = \frac{I_A}{\tau_A} - \frac{I_S}{\tau_S},$$

$$\frac{dR}{dt} = \frac{I_S}{\tau_S},$$

$$\frac{dD}{dt} = \beta_A m_A I_A + \beta_S m_S I_S - \frac{D}{\tau_D},$$

$$\frac{dF}{dt} = \frac{D}{\tau_D} - \frac{F}{\tau_F}.$$

*$R_0$ calculation*

Given a population size of $N$ (large, $\gg R_0$), all susceptible and the introduction of one exposed individual:

Total droplet inoculum produced

$$D_{\text{Total}} = \beta_A \tau_A + \beta_S \tau_S.$$

Total infections caused by droplet ($D$) and fomite ($F$) inoculum

$$I_D = \beta_D N D_{\text{Total}} \tau_D,$$

$$I_F = \beta_F N F_{\text{Total}} \tau_F.$$

Thus $R_0$, i.e. total infections caused by the initial introduction (noting $D_{\text{Total}} = F_{\text{Total}}$)

$$R_0 = (\beta_D \tau_D + \beta_F \tau_F) D_{\text{Total}} N,$$

$$R_0 = (\beta_D \tau_D + \beta_F \tau_F)(\beta_A \tau_A + \beta_S \tau_S) N.$$

Given the proportion of $R_0$ due to droplets, $\mu = \frac{\beta_D \tau_D}{(\beta_D \tau_D + \beta_F \tau_F)}$, we obtain a relation between the droplet and fomite spread rates ($\mu \neq 1$)

$$\beta_D = \left( \frac{\mu}{1 - \mu} \right) \frac{\tau_F}{\tau_D} \beta_F$$

and so

$$\beta_D = \left( \frac{\mu}{\beta_A \tau_A + \beta_S \tau_S} \right) \frac{R_0}{N \tau_D}, \beta_F = \left( \frac{1 - \mu}{\beta_A \tau_A + \beta_S \tau_S} \right) \frac{R_0}{N \tau_F}.$$

A summary of the model parameters and default values for the modified SIR compartmental model with free-living inoculum is given in table 1.

## (ii) Additional assumptions and knowledge gaps

A key assumption of the model is that those people who have been infected and have recovered have immunity to further infections. At present, there are no experimental data to validate this assumption [48] and it is widely accepted that the four related, seasonal coronaviruses, responsible for up to 30% of common colds, cause illness repeatedly, even though people have been exposed to them throughout their lives.

We also make assumptions and simplifications about the effectiveness of facemasks because there is a wide range of possible designs [49]. In general, however, the important fact here is that inoculum release is principally through the mouth, which the facemask covers effectively, so this is a key point is estimating the facemasks' efficacy at catching exhaled inoculum. Evidence from the literature shows that the filtration efficiency of different cotton-fabric facemasks varies between 43% and 94% in controlling the passage of bacteria [47], which travel in moisture droplets in the same way as viruses. Droplet-blocking efficiency of fabric samples was shown to be 90–98% for 100% cotton T-shirt, dishcloth and silk shirt, which was as high as for fabric material used for the production of a three-layered commercial medical mask [50]. More recent research showed that surgical facemasks significantly reduced the detection of influenza virus RNA in respiratory droplets and coronavirus RNA in airborne droplets and it was concluded that surgical facemasks could prevent coronavirus transmission from symptomatic individuals [51]. In addition, facemasks capture the larger droplets most effectively, and these carry most of the virus load (a droplet with twice the diameter has eight times the weight and virus content). Further qualitative evidence of the effectiveness of facemasks at catching exhaled infection agents is the universal acceptance of the need for surgeons to wear a facemask when operating to avoid the infection of the wound on which they are working. We assume, therefore, that facemask effectiveness in exhalation is probably well above 50% on average, but probably poorer on inhalation; we justify this in terms of the difference in dynamics between airflow coming out from or going into an orifice (nose or mouth). This can be illustrated by the ability to blow out

**Table 1.** Model parameters and default values for the modified SIR compartment model with free-living inoculum. Note that our inferences, choices of parameters and population sizes do not relate to healthcare environments, as would be found in hospitals, where inoculum levels may be extremely high and personnel already wear appropriate PPE.

| parameter | description | default value | source |
|---|---|---|---|
| $N$ | population size | 60 million | approximate mainland GB population size |
| $R_0$ | basic reproductive rate | 4 | [30] |
| $\beta_A$ | inoculum release rate of asymptomatic infectious individuals | 2.71 unit inoculum per day per capita | relative to $\beta_S$ [39] |
| $\beta_S$ | inoculum release rate of symptomatic infectious individuals | 1 unit inoculum per day per capita | arbitrarily defined, without loss of generality |
| $\beta_D$ | infection rate due to droplet inoculum | $4.46 \times 10^{-5}$ per unit inoculum per day | fitted subject to default values of $R_0$, $\mu$ and $N$ |
| $\beta_F$ | infection rate due to fomite inoculum | $2.58 \times 10^{-9}$ per unit inoculum per day | fitted subject to default values of $R_0$, $\mu$ and $N$ |
| $m_A$ | reduction in inoculum release rate of asymptomatic individuals for mask wearers | 0.5 | arbitrarily set in the absence of detailed data on individual-based transmission; consistent with lower ranges quoted by Furuhashi [47]; van der Sande et al. [25] |
| $m_S$ | reduction in inoculum release rate of symptomatic individuals of mask wearers | 0.5 | arbitrarily set in the absence of detailed data on individual-based transmission; consistent with lower ranges quoted by Furuhashi [47]; van der Sande et al. [25] |
| $m_D$ | reduction in inoculum infection rate due to droplet inoculum for mask wearers | 0.5 | arbitrarily set in the absence of detailed data on individual-based transmission |
| $m_F$ | reduction in inoculum infection rate due to fomite inoculum for mask wearers | 1.0 | arbitrarily set in the absence of detailed data on individual-based transmission |
| $q$ | reduction of transmission rates from lock-down of the population | 0.5 | [30] showing $R_e$ dropped from $\sim$4 before to $\sim$1 after lock-down |
| $\tau_E$ | average duration between infection and onset of asymptomatic infectiousness | 3.8 d | [39] |
| $\tau_A$ | average duration between onset of asymptomatic infectiousness and first symptoms | 1.2 d | [39] |
| $\tau_S$ | average duration between first symptoms and end of infectiousness | 3.2 d | [39] |
| $\tau_D$ | average lifespan of droplet inoculum before deposition | 10 s | [44–46] |
| $\tau_F$ | average lifespan of fomite inoculum before loss of viability | 48 h | [43] |
| $\mu$ | assumed proportion of infections due to droplet inoculum in the absence of masks or other forms of control | 0.5 | arbitrarily set in the absence of detailed data on individual-based transmission |

a lit match at arm's length compared with the impossibility of putting it out by sucking in air. Exhaled air, therefore, is likely to go mostly through the fabric, whereas inhaled air is much more likely to enter around the sides of a facemask, if it is unsealed.

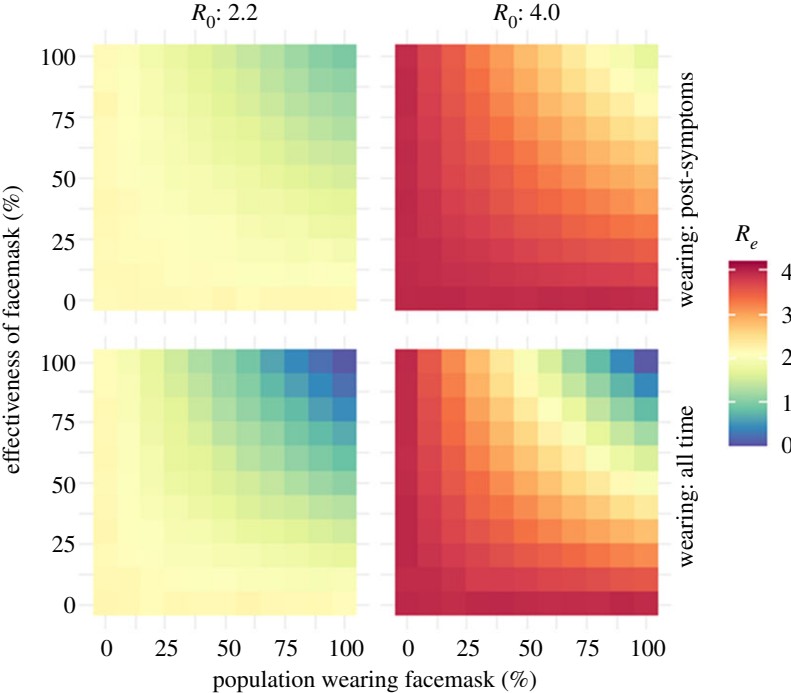

**Figure 3.** Heat maps of the effective reproduction number ($R_e$) as a function of control parameters for two values of $R_0$. Even when $R_0$ is 4.0, the best outcomes are achieved when masks are worn all the time, a high proportion of the population wear them and their efficacy is high.

## 3. Results

### (a) Model 1: branching process transmission

We explore how the effective reproductive number ($R_e$), i.e. the average number of infections generated by an infectious individual in the population, changes under a range of possible values for the proportion of a population wearing facemasks and the efficacy of the facemasks. Figures 3 and 4 show results for simulations with the branching process model run initiated with 100 infectious individuals in generation 1 and the proportion of the population wearing a facemask after generation 3. We distinguish between two scenarios: facemasks worn after symptoms are expressed and facemasks worn all the time. By 'all the time', we mean compliance with normal facemask procedures when in public [52], irrespective of whether or not COVID-19 symptoms are being expressed.

Even when the initial reproductive number ($R_0$) is 4.0, our analyses show that the best reductions in the $R_e$ occur when facemasks are worn all the time, by a high proportion of the population and their efficacy is high (figure 3). It is also possible (subject to the simplifying assumptions) for $R_e$ to be brought below 1, when the public wear effective facemasks all of the time (rather than only starting when COVID-19 symptoms appear), leading to the epidemic dying out. These exploratory analyses from this model suggest that there are regions of parameter space (proportion of population wearing facemasks and effectiveness of the facemasks in reducing transmission) in which this intervention strategy could be effective. We now examine the dynamics in more detail, with a more mechanistic model, albeit in which there is still uncertainty over some parameter values.

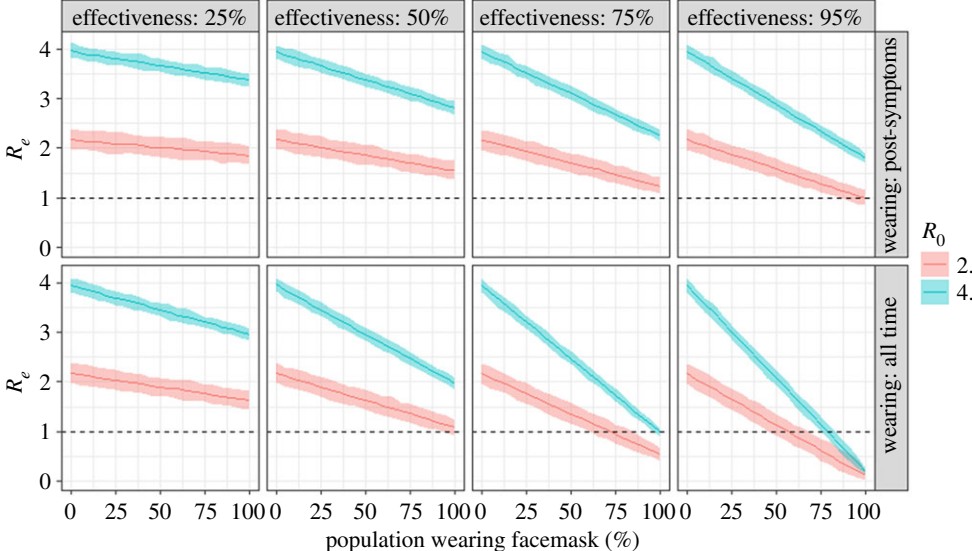

**Figure 4.** The effect on $R_e$ of the proportion of the population wearing a facemask. The solid line is the mean and the shaded area shows the 95% confidence interval. By the public wearing an effective facemask all of the time (rather than just starting when COVID-19 symptoms appear), the $R_e$ can be brought below 1, leading to the pandemic dying out.

## (b) Model 2: the adapted susceptible–infected–removed compartment model

This model allows us to investigate the interactions between lock-down periods and various percentages of facemask adoption by the public. Here, we examined the effects of changing the proportion of people wearing masks (0%, 25%, 50%, 100%), when there are successive periods of lock-down. Our focus here is on the potential for reducing SARS-CoV-2 transmission by facemasks and we do not consider any other intervention besides population lock-down.

We first examine the epidemic dynamics, for different proportions of facemask wearers, when facemask wearing is initiated at the beginning of the epidemic. The epidemic takes off exponentially and is only slowed by the imposition of the first lock-down period, which we assume reduces overall transmission of the virus (table 1 and figure 5a). A second wave begins after the first lock-down is lifted, which is suppressed by the second lock-down period. By the third lock-down period, under this scenario, everyone has become infected and the epidemic dies down. This does assume that infected/recovered people have become immune/resistant. With 25% facemask adoption, the initial peak is flattened, but the second wave is more pronounced (figure 5b). At 50% adoption, secondary and tertiary disease peaks occur in the second and third lock-down periods (figure 5c). The benefits of the additional reduction in transmission rate effected by facemasks are fairly equally divided between facemask and non-facemask wearers (figure 5c). When 100% of the public wear facemasks, disease spread is greatly diminished and the total numbers of 'removed' are substantially lower. Here, we consider an 18 month time scale, either until the introduction of a vaccine or to the point at which sustained lock-down control efforts are not feasible.

We next relax the assumption that facemask wearing begins from early in the epidemic when there are just 100 detected cases (as in figure 5) and investigate the later adoption at 30, 60, 90 and 120 days. While later adoption leads to increases in the numbers of individuals who become infected, even when implemented at 120 days after the initiation of the epidemic (figure 6), 100% adoption of facemasks by the public (under the current assumptions: table 1 and figure 6d) stops the occurrence of further COVID-19 epidemic waves. We also note that there are benefits to employing facemasks even when $R_e$ has fallen below 1. For example, the number of active cases drops off much faster in figure 6b, where the adoption of facemasks occurs shortly after the peak

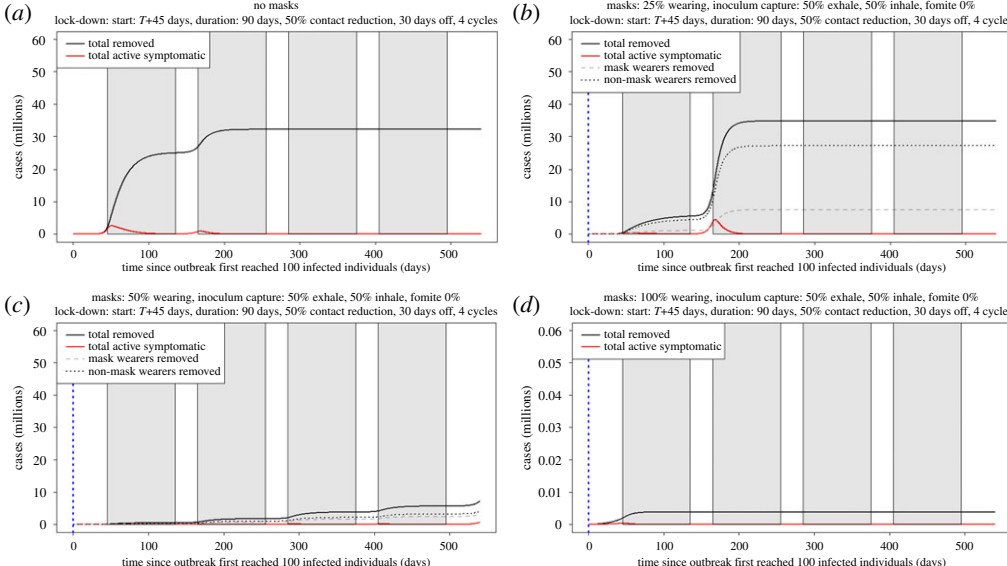

**Figure 5.** The vertical, blue, dotted line indicates the time at which facemask wearing is adopted. (*a*) With no facemasks, the disease progress curve increases exponentially before periods 2 and 3 and is only slowed by the imposition of the first lock-down period. When this ends, a second wave begins, which is suppressed by the third lock-down period. By the fourth lock-down period, everyone has become infected and the epidemic dies down. This does assume that infected/recovered people have become immune/resistant. (*b*) With 25% facemask adoption, the initial three peaks are flattened, but a second, larger wave appears in the fourth lock-down period. There are clear benefits to facemask wearers, compared with non-wearers. (*c*) At 50% adoption, the disease progress curve does not take-off until after the fourth lock-down period. (*d*) When 100% of the public wear facemasks, disease spread is greatly diminished and the total numbers of 'removed' is much lower (note the different *y*-axis scale for *d*).

in active cases is reached, than in figure 6*d*, where facemasks are not adopted until later. This faster drop in active cases could allow for an earlier lifting of lock-down. We note that, when the number of cases is reduced to a sufficiently low level, other forms of control such as contact tracing become more feasible.

For completeness, we analysed the effects of facemask use in the absence of lock-down or other mitigation procedures (figure 7). We note that, in scenarios 5*c* and 5*d*, the epidemic has not infected enough of the population to reach herd immunity in the absence of periodic lock-downs. The long-term dynamics of these outbreaks after the relaxation of lock-down are examined in figure 6. The default parameters remain as in table 1 and salient parameters are repeated in figure 7. It is clear that, consistent with an epidemic with an $R_0$ value of approximately 4 [30], the epidemic increases exponentially, leading to very high levels of infection. Here, as elsewhere, we assume that infection confers immunity, though that assumption can easily be relaxed by expanding the framework with free-living inoculum to incorporate a SIRS model, where after some time removed individuals can become susceptible again. Adoption of facemask wearing by 25% of the population decreases the level of infection in the population (figure 7*b*). The effects increase with greater adoption (figure 7*c,d*). At 100% adoption (figure 7*d*), the disease progress curve is flattened significantly and the total number of individuals infected is reduced. We note that 100% facemask adoption without lock-down achieves a greater reduction in the final size of epidemic, a lower 'total removed' and a lower peak of active cases than lock-down without facemasks (figure 5*a*). These results are striking in that the benefits accrue to the facemask wearer as well as to the population as a whole. We have assumed a reduction each in inhalation and exhalation of 50% of droplet inoculum compared with no facemask.

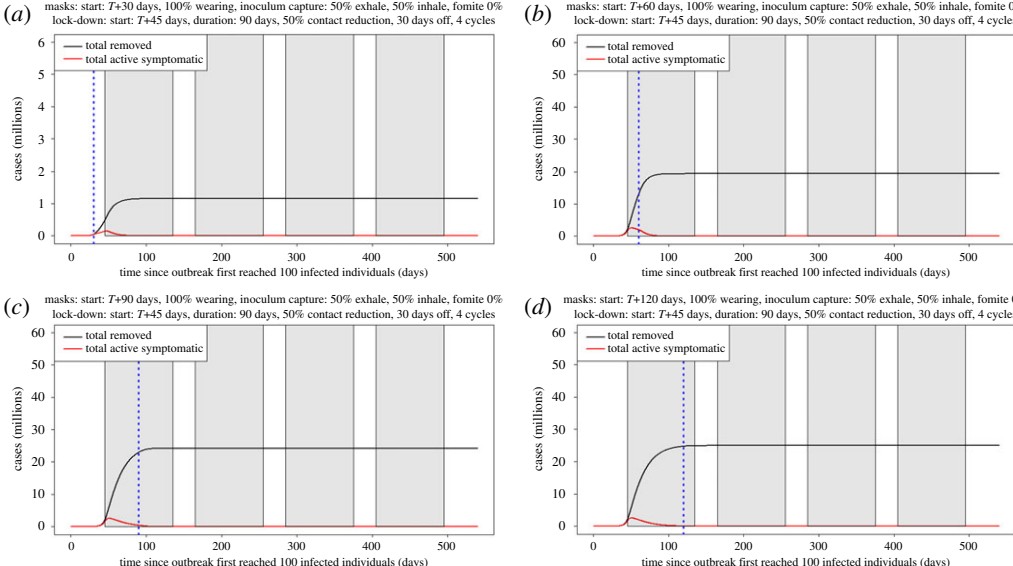

**Figure 6.** Effect of changing start date for facemask adoption, shown by the blue dashed line. (*a*–*d*) $T + 30d$, $T + 60d$, $T + 90d$, $T + 120d$, respectively. The first lock-down period started at $T + 90d$. Earlier adoption of facemasks (*a*) has a different *y*-axis scale, because the end values of total removed are significantly lower. Even when implemented at $T+120$ days, 100% adoption of facemasks by the public stops the occurrence of additional COVID-19 pandemic waves.

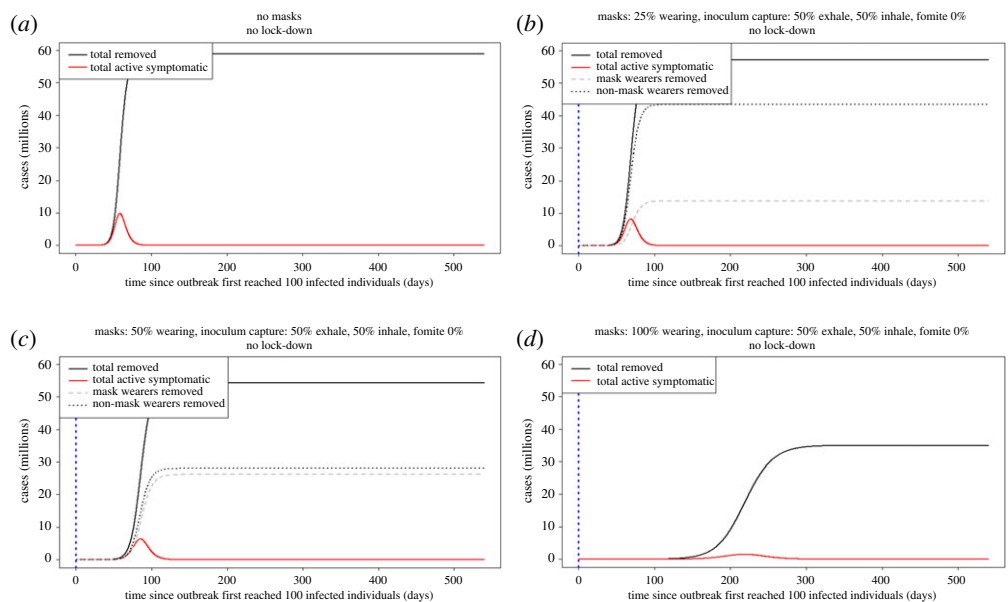

**Figure 7.** Effects of facemask adoption in the absence of lock-down periods, using the same facemask-wearing proportions as in figure 6. The vertical, blue dotted line indicates the time at which facemask wearing is adopted. (*a*) Default epidemic dynamics in the absence of any intervention. (*b*) Twenty-five per cent facemask wearers in the population slows the overall epidemic progression, but provides minimal reduction in final size. (*c*) Fifty per cent adoption of facemasks further slows epidemic progression and slightly decreases final size, with benefits distributed fairly evenly between mask and non-mask wearers. (*d*) One hundred per cent adoption with 50% reduction in inhalation and exhalation leads to flattening and delay of the disease progress curve and the total number of individuals infected in the population is reduced (see table 1 for default parameters).

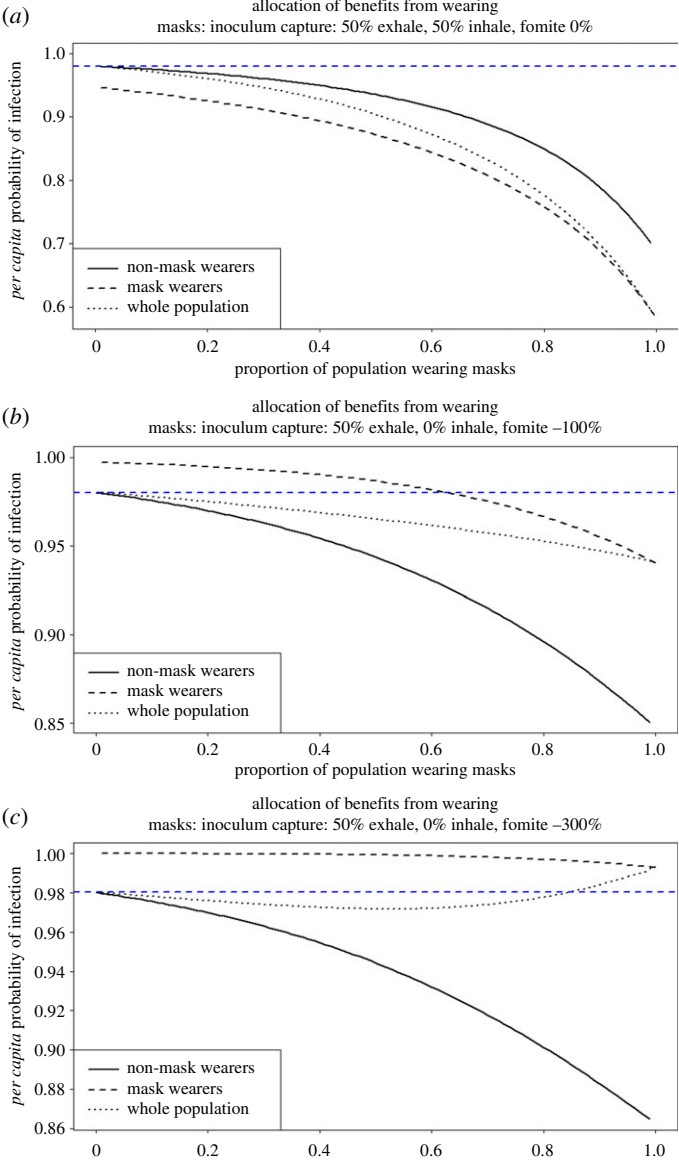

**Figure 8.** The *per capita* probability of infection in relation to the proportion of the population wearing facemasks and the negative effects of fomite infection, owing to poor compliance. The blue dashed line represents the individual probability of infection when no one wears a facemask. (*a*) When the fomite parameter is at 0% and droplet exhalation and inhalation are both reduced by 50% for facemask wearers (good compliance and facemask), facemask wearers benefit more than non-facemask wearers and the probability of infection drops with increasing proportion of the population wearing facemasks, and is always lower for both facemask wearers and non-wearers than if nobody wears a mask. (*b*) When facemask wearing incurs a susceptibility penalty, with the fomite susceptibility increased by 100% and a 50% reduction in droplet exhalation, but no effect on droplet inhalation (bad compliance and facemask), there is a net benefit to the population, but the mask wearers are personally worse off until about 65% of the population is wearing them, when even the facemask wearers become better off than if nobody wears facemasks. (*c*) When compliance and facemask usage are very bad (fomite susceptibility increase of 300% and 50% droplet exhalation reduction), non-facemask wearers are at an advantage but, counterintuitively, until about 80% facemask adoption, there is still a small net benefit to the population as a whole, although a larger benefit to non-mask wearers. Note the different axis scales in each subplot, e.g. the facemasks in (*a*) are able to deliver a 40% reduction in population-level risk, while the facemasks in (*b*,*c*) are only able to deliver up to a 15% reduction, to a portion of the population.

It is also instructive to consider the *per capita* probability of infection in relation to the proportion of the population wearing facemasks and the negative effects of fomite infection, owing to poor compliance and ineffective facemasks. When the fomite parameter for infection is at 0% (good compliance and facemask), facemask wearers benefit more than non-facemask wearers and the probability of infection drops with increasing proportion of the population wearing facemasks. As the proportion of mask wearers increases, the benefits to both facemask wearers and non-facemask wearers increase (figure 8*a*). We see that if facemask wearing were to incur a susceptibility penalty—with the fomite parameter increased by 100% and no protection against droplet inhalation, while retaining the 50% reduction in droplet exhalation—there is still a net benefit to the population. In this scenario, the mask wearers are personally worse off until about 65% of the population is wearing masks, at which point even the facemask wearers become better off than if nobody in the population was wearing facemasks (figure 8*b*). When compliance and facemask wearing are very bad (300% increase in fomite susceptibility with only the 50% reduction in droplet exhalation), non-mask wearers are at an advantage, but a highly counterintuitive finding is apparent in that, until about 80% facemask adoption, there is still a net benefit to the population as a whole, and a greater benefit for non-mask wearers. This non-intuitive result is due to the fact that, while an individual is made much more susceptible, this is more than counteracted by the reduced infectiousness of many individuals. These facemask wearers are more likely to contract the disease, but less likely to infect others (particularly non-facemask wearers) once infected. When most of the population is wearing these facemasks, the effect of reduced infectiousness is outweighed by the fact that the majority of the population is much more susceptible. In the unlikely eventuality of facemasks with this property, it would be advisable to preferentially issue these masks to individuals less vulnerable to severe complications, leaving more vulnerable individuals unmasked in order to maximize overall benefits, while still achieving a lower level of infection within the population.

## 4. Discussion

The emergence of the COVID-19 pandemic has stimulated a search for possible interventions, such as SARS-CoV-2 vaccines, but these may not be effective or become available in the near future [53]. There has also been a lack of clarity in the thinking surrounding the potential benefits of population-level adoption of facemasks, which could provide a cheap and effective means of managing COVID-19 epidemics in high-, middle- and low-income countries.

Previous work used a similar mathematical model-based approach to estimate the relative contributions of the four pathways to infection risk in the context of a person attending a bed-ridden family member ill with influenza [54]. Our models, however, concern SARS-CoV-2 and are focused on the population consequences of lock-down periods and facemask adoption. Our conclusions are similar, however, in that, given the current limited information on dose and infectivity via different exposure pathways, non-pharmaceutical interventions should simultaneously address potential exposure via face-touching behaviour and inhalation of virus particles. A separate study also recently analysed the effectiveness of facemasks using a compartmental, ordinary differential equation model, albeit in a more conventional form, without the incorporation of free-living inoculum [55]. Similarly, they concluded that facemasks were valuable in reducing virus transmission. They also demonstrated that the relative benefit of facemask use was greater when adopted earlier [55]. Neither Nicas & Jones [54] nor Eikenberry *et al*. [55] considered facemasks in combination with lock-down periods.

We first used a relatively simple mathematical model to ascertain the potential impacts of facemasks, given the transmission characteristics of SARS-CoV-2. Then, we used a modified SIR model with free-living inoculum that enables distinction between SARS-CoV-2 particle shedding and uptake/infection to integrate cycles of lock-down periods interspersed with release of the public. Our intention here is not to reproduce detailed, realistic, individual-based simulation models [56] but rather to introduce and analyse a simple mechanistic model that can be used to inform understanding of fundamental dynamics. We use available literature as well as plausible

assumptions to parametrize our models. It is understood, however, that our estimated outcomes are influenced by these parameter ranges and values.

Our mechanistic, modified SIR model can be used to analyse any intervention that affects transmission rates, with the advantages of separating droplet from fomite components. This model could also readily be made more realistic and complex, by considering and including linked subpopulations within a metapopulation, stochastic processes and allowing for uncertainty by sampling parameters from posterior or other distributions.

Our approach is to accept that, with a new disease, it is impossible to get accurate experimental evidence for potential control interventions, but that this problem can be approached by using mathematical modelling tools to provide a framework to aid rational decision-making. For completeness and objectivity, we also model a scenario where facemask adoption had negative effects. We do consider this scenario unlikely, however, because countries where facemask wearing is mandatory are currently experiencing relatively low numbers of COVID-19 cases and deaths [57,58].

Both of our models show that, under a wide range of plausible parameter conditions, facemask use by the public could significantly reduce the rate of COVID-19 spread, prevent further disease waves and allow less stringent lock-down regimes. The effect is greatest when 100% of the public wear facemasks. It follows that the adoption of this simple technology ought to be re-evaluated in countries where facemask use is not being encouraged. Within the parameter regimes tested, the models also show that, if COVID-19 is to be controlled or eradicated, early lock-down combined with facemask adoption by close to 100% by the public needs to occur. This, of course, does not exclude the implementation of other management interventions, such as widespread testing and contact tracing.

The detailed conclusions that can be drawn from the branching process, transmission model are:

  (i) With a COVID-19 $R_0$ of 2.2 and for scenarios where facemasks were worn only after onset of symptoms, the median $R_0$ fell below 1, if at least 95% of the population wore facemasks (with an efficacy similar to that of N95 respirators designed to achieve a very close facial fit and highly efficient filtration of 0.3 µm airborne particles).
 (ii) When the COVID-19 $R_0$ was higher (4.0), wearing a facemask after the onset of symptoms still decreased the median $R_e$ to just below 2, when there was high adoption and efficacy of facemasks.
(iii) With a policy that all individuals must wear a facemask all of the time, a median effective COVID-19 $R_0$ of below 1 could be reached, even with facemask effectiveness of 50% (for $R_0 = 2.2$) or facemask effectiveness of 75% (for $R_0 = 4$).

Our analyses from this model are consistent with the conclusions of practical studies on the use of professional and home-made facemasks to reduce exposure to respiratory infections among the public, where it was concluded that any type of general mask use is likely to decrease viral exposure and infection risk on a population level, in spite of imperfect fit and/or adherence [25]. Other studies on respiratory virus transmission have concluded that, for compliant users, facemasks were highly efficacious at preventing spread [59]. In a pandemic situation as now exists, compliance is affected by perception of risk, so we would expect compliance to be high.

The detailed conclusions from the modified SIR model are consistent with those of the first model and are that:

 (iv) Lock-down periods alone do not prevent the occurrence of secondary and tertiary waves of the pandemic occurring and these may be larger than the initial wave (figures 5 and 6).
  (v) If lock-down periods are combined with 100% adoption of facemask use by the public, the initial disease progress peak is dramatically flattened and delayed and subsequent waves are prevented (figures 5c and 7a).

(vi) At the time of writing, facemask or face-covering use by the public has not been recommended by the UK, apart from in Scotland. We considered, therefore, the effects of varying the time of facemask adoption by the public and show that, even if facemask use began immediately while in the first lock-down period, major benefits would still be accrued by preventing the occurrence of further COVID-19 disease waves (figure 6a–c).

(vii) We consider a scenario, for completeness, where there are no lock-down periods and facemask use has negative effects on the wearer (figure 8b,c). This is clearly the worst case but we consider that, in practice, negative effects of facemask adoption are improbable, because countries where facemask use is mandatory have relatively low COVID-19 spread and numbers of deaths. Even here, however, there is the highly counterintuitive outcome that, although the facemasks were bad for the individuals wearing them (because of extremely poor compliance issues and a bad facemask design), people wearing them still provided a net benefit to the population as a whole. Here, we have analysed a range of possible values for the enhanced risk from fomite inoculum to facemask wearers. With currently available data, it is not possible to identify which, if any, of these values is most likely. Further experimental work is required to address this important aspect of facemask use.

(viii) We examined the effects of different rates of facemask adoption alone (with no lock-down periods) by the public and show that, at lower levels of adoption, there are real benefits to facemask wearers. This difference is reduced as the per cent facemask adoption increases to 50%. At 100% adoption, the disease progress curve is flattened and delayed significantly, as well as the total removed from the population being reduced. There is, therefore, a clear incentive for people to adopt facemask wearing (figures 7 and 8a). This is a similar conclusion to that reached by another modelling study that concluded that facemask use by the public is potentially of high value in curtailing community transmission and the burden of the pandemic [55].

(ix) A combination of facemask wearing and lock-down periods implemented together is indicated to provide a better solution to the COVID-19 pandemic than either in isolation. The models indicate that a combined mitigation strategy is required to reduce the transmission of the pathogen to flatten and delay the disease progress curves and to prevent repeated waves of infection. Here and elsewhere, our results are subject to the assumptions in our models, including some arbitrary but reasonable assumptions about key parameters, given the current state of knowledge. The effects of control measures observed in the results presented here with $R_0 = 4.0$ are greatly enhanced with a lower $R_0$.

When a new disease to humanity, caused by a pathogen, first appears and spreads, there is an unavoidable lack of information on which to base rational control-intervention decisions. An obvious knowledge gap is the lack of accurate data on the period when individuals are non-symptomatic, but infectious. These data would clearly help with contact tracing, in terms of the duration of the historical search-period required. Also, information on the size of the infectious dose for SARS-CoV-2 is lacking, but research on SARS-CoV (the 2003 virus) showed that between 43 and 280 individual virus particles had to enter the human body in order to start an infection [60]. Even home-made masks, washed in soapy water after each use, therefore, should reduce fomite build-up and droplet-mediated airborne transmission. Surgical-grade masks, however, would probably be required to reduce transmission by droplets containing viral inoculum, as was reported for the influenza virus [61].

In the current COVID-19 pandemic, one solution to this dearth of information has been to advocate implementation of the precautionary principle on interventions, which can be defined as 'a strategy for approaching issues of potential harm, when extensive scientific knowledge on the matter is lacking'. Evidence on the efficacy and acceptability, or otherwise, of facemasks to combat respiratory disease spread, for example, is sparse and contested [62], but deaths in many countries are rising steeply, with little time as yet to collect appropriate and rigorous scientific data. The

precautionary principle approach would be to encourage populations to adopt facemask wearing immediately, given that facemasks are considered essential PPE for front-line healthcare workers and that 'we have little to lose and potentially something to gain from this measure' [63].

Our modelling framework highlights the urgent need to translate individual-based research on coughing, sneezing and exhaling into SARS-CoV-2 transmission rates, as shown in figure 2. One way to do this is to identify viral loads of SARS-CoV-2 that are required for infection and also to quantify the dispersal of particulate inoculum. That would also have a bearing on social distancing advice where, to date, the identification of 2 m is arguably arbitrary. In a study investigating environmental contamination by SARS-CoV-2 from a symptomatic patient wearing an N95 facemask, no SARS-CoV-2 particles were detected on the front surface of the facemasks worn by the study's physicians [65]. Other recent evidence on facemasks for reducing SARS-CoV-2 transmission is provided by an epidemiological investigation where a patient transmitted COVID-19 to five people in one vehicle when he did not wear a facemask. In a later journey, no one was infected in the second vehicle when he wore a facemask [5].

Although our modelling framework demonstrates that, under certain conditions, facemask adoption by entire populations would have a significant impact on reducing COVID-19 spread, there are additional human factors and obstacles that may prevent the implementation of this policy or a directive being issued at a governmental level. The most important of these is probably the perceived lack of availability of efficient facemasks (N95 respirators) and the view that these should and must be reserved for front-line medical workers.

In emergency situations such as this where there are acute PPE shortages, however, it would be pragmatic and acceptable for people to improvise and fabricate their own solutions to the problem [25,49,66]. The natural mechanics of filtration are that larger droplets are captured more effectively. So, it can safely be assumed that droplets in the 1 µm plus range will be almost completely eliminated by such an informally made mask. This is very important because a 2 µm droplet has a thousand times the mass of a 200 nm droplet, and a 20 µm droplet has a million times the mass of a 200 nm droplet, the virus load being proportional to the mass. The larger the droplets, the more important it is to capture them, and even a home-made mask will do this very well. There are also experimental data, for instance, that show that home-made facemasks consisting of one facial tissue (inner layer on the face) and two kitchen paper towels as the outer layers achieved over 90% of the function of surgical mask in terms of filtration of 20–200 nm droplets [25]. These facemasks are also disposable, so would clearly provide a pragmatic solution to the problem, and there are many sites now on social media that provide clear instructions on facemask making and safe use [49].

Our analysis indicates that a high proportion of the population would need to wear facemasks to achieve reasonable impact of the intervention. In Hong Kong, 99% of survey respondents reported wearing facemasks when outside of their home [67]. Another human factor that may reduce facemask adoption in Western countries is cultural, because the use of facemasks is not common in public, or there is an implication that the facemask wearer considers others as a threat. In the current emergency, however, it is necessary to change this view, which could be achieved if the message conveyed by a facemask was 'my facemask protects you, your facemask protects me'. Indeed, it is probable that making facemasks into fashion items may be another route to changing the culture surrounding facemask use in public. A further positive effect from this cultural change would be to reinforce the message that it is necessary to keep to a safe distance from one another. This educational message could be conveyed easily by the government and popular press.

While our mathematical models indicate the need for improved parameter estimates, especially for $\beta_i$ and $m_i$, but also for the intrinsic epidemiological parameters ($\tau_i$) and decay rates for droplet inoculum and fomite, they have also indicated some important interactions with respect to timing of interventions in relation to the periodicity imposed on the dynamics by periods of lock-down. Our exploratory analyses when modelling the initiation of facemask adoption before, during and after an initial lock-down period indicate that the precise balance of infecteds and susceptibles upon release from lock-down can have profound, counterintuitive consequences for the occurrence of subsequent waves and the ultimate numbers of infected

individuals in the population. This and further analyses of the benefits to facemask and non-facemask wearers are the subject of further research.

Potential extensions to the modelling framework presented here include the extension to a metapopulation, spatially partitioning areas of high and low contact risk, as well as the incorporation of human behavioural shifts dependent on an individual's perceived risk of disease, adoption fatigue or inconsistent adoption.

Despite the potential for facemasks to reduce SARS-CoV-2 transmission, there does not appear to be any focus on investing efforts in properly designed studies on facemasks, or evaluating large populations including 'at risk' patients and in a variety of communities. We argue that these are required urgently, and our modelling framework can readily be adapted to incorporate any new data that become available.

In summary, our modelling analyses provide support for the immediate, universal adoption of facemasks by the public, similar to what has been done in Taiwan, for example, where production will soon reach 13 million facemasks daily, with well-developed plans for N95 respirator production in the pipeline [68]. Our analyses indicate that actions to facilitate this in the UK should include clear instructions on the fabrication and safe use of home-made masks, as well as accompanying governmental policies to increase swiftly the availability of medical standard surgical, or N95 respirators, to the public.

Data accessibility. The code for models 1 and 2 is available at https://github.com/camepidem/COVID-19_PRSA.

Authors' contributions. The original high-level question concerning the effectiveness of facemasks at population scales was proposed by J.C. and M.B. to C.A.G., R.O.J.H.S. and R.R. RR designed, implemented and analysed the branching process model while R.O.J.H.S. did so for the SECIR-DF compartmental model in discussion with C.A.G. J.C., C.A.G., R.O.J.H.S. and R.R. reviewed and discussed the results, and drafted the paper. M.B. reviewed the manuscript and gave technical inputs on mask designs. All authors reviewed and agreed the final form of the manuscript and agree to be held accountable for the work performed therein.

Competing interests. We declare we have no competing interests.

Funding. R.O.J.H.S., R.R. and C.A.G. acknowledge support from the Epidemiology and Modelling Group, Cambridge, UK. M.B. acknowledges support from the Wolfson Centre, University of Greenwich, UK. J.C. acknowledges support from the Natural Resources Institute, University of Greenwich, UK.

Acknowledgements. We are grateful to Dr Sharon van Brunschot and Theodore Bradley for percipient suggestions for the draft article and Dr Fiona Hansell for consultation on medical terminology. We also thank the two reviewers for helpful comments.

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
