## [Reviewer comments · Proceedings. Mathematical, Physical, and Engineering Sciences]

Review History

RSPA-2020-0376.R0 (Original submission)

Review form: Referee 1

Is the manuscript an original and important contribution to its field?

Excellent

Is the paper of sufficient general interest?

Excellent

Is the overall quality of the paper suitable?

Excellent

Can the paper be shortened without overall detriment to the main message?

Yes

Do you think some of the material would be more appropriate as an electronic appendix?

No

Do you have any ethical concerns with this paper?

No

Recommendation?

Accept with minor revision (please list in comments)

Comments to the Author(s)

This paper addresses an important topic in a timely manner. It presents models that can be used to assess the effectiveness of facemasks in managing COVID-19 under various scenarios, including lockdowns. The modelling is appropriate and has been carefully conducted on the basis of reasonable assumptions. The paper is well written and mostly very clear. The paper could well have an influence on government policy. I recommend publication with only very minor changes.

I have inserted comments with suggestions for minor re-wording in the pdf file of the paper. The commented MS is included with this review.

For the non-specialist reader it would help to have a clear definition of terms such as R_e and R_o when they are first introduced. Then, once the terms are introduced, they should be used consistently, rather than, for example, saying "effective reproductive number".

Review form: Referee 2**Is the manuscript an original and important contribution to its field?**

Good

Is the paper of sufficient general interest?

Excellent

Is the overall quality of the paper suitable?

Good

Do you have any ethical concerns with this paper?

No

Recommendation?

Accept with minor revision (please list in comments)

Comments to the Author(s)

This is an interesting and timely piece on the added benefits of wearing facemasks as an addition to a lockdown on COVID reproduction rate. The manuscript is well written and clear. It employs two models to make conclusions (a branching model and SIR model).

The authors use facemask effectiveness in influenza outbreaks as an example comparator. Is there any evidence for their use in Tuberculosis, especially in LMIC settings.

When analysing the data, it may be easier to be clarify:

(1)The model focuses at mask level rather than the droplet dynamic level

(2)What is the distribution of the mask-status-outcome relationship? Unlikely to be linear, is it discrete distribution?

(3) Should there be a measure of time of mask worn in a contagious area and probabilistic outcome

(4) Can there be a measure of efficacy of mask worn? An assumption was made here but can this be updated?

(5) As this work is focussing on a lockdown+mask, what lessons can we discern from masks without lock down

(6) There is good modelling evidence of bringing down R from >1 to <1 , however is there any clarity of the use of the mask and its efficacy when R exists below 1 and subsequent results from the model here?

Mask wearing is unlikely to be constant over time – delayed adoption, adoption fatigue etc. Understanding the sensitivity of the models to these human factors would be helpful to determine the applicability of these findings. Similarly, compliance may be inversely correlated with incidence – e.g. at day 300, if there have been very few cases recently, why should I keep wearing a mask?

Modelling

For the SIR model, R_0 is taken to be 4, yet earlier results show R_e is highly dependent on the chosen value of R_0 . It may be useful to include R_0 of 2.2 in the SIR model demonstrate the potential range of benefits arising from mask wearing.

The impact of fomite transmission in the model introduced some counterintuitive findings with non-facemask wearers benefitting more than those wearing a mask. The likelihood of each of the 0, 100 and 300% conditions would be very helpful to understand the most likely scenario encountered, if possible.

Consistency in y-axis ranges in Figure 8 would be helpful for comparison between panels.

Decision letter (RSPA-2020-0376.R0)

15-May-2020

Dear Dr Stutt,

On behalf of the Editor, I am pleased to inform you that your Manuscript RSPA-2020-0376 entitled "A modelling framework to assess the likely effectiveness of facemasks in combination with 'lock-down' in managing the COVID-19 pandemic" has been accepted for publication subject to minor revisions in Proceedings A. Please find the referees' comments below.

The reviewer(s) have recommended publication, but also suggest some minor revisions to your manuscript. Therefore, I invite you to respond to the reviewer(s)' comments and revise your manuscript. Please note that we have a strict upper limit of 28 pages for each paper. Please endeavour to incorporate any revisions while keeping the paper within journal limits. Your paper has been ESTIMATED to be 23 pages. If you have any questions, please do get in touch.

It is a condition of publication that you submit the revised version of your manuscript within 7 days. If you do not think you will be able to meet this date please let me know in advance of the due date.

To revise your manuscript, log into <https://mc.manuscriptcentral.com/prsa> and enter your Author Centre, where you will find your manuscript title listed under "Manuscripts with Decisions." Under "Actions," click on "Create a Revision." Your manuscript number has been appended to denote a revision.

You will be unable to make your revisions on the originally submitted version of the manuscript. Instead, revise your manuscript and upload a new version through your Author Centre.

IMPORTANT: Your original files are available to you when you upload your revised manuscript. Please delete any redundant files before completing the submission process.

In addition to addressing all of the reviewers' and editor's comments, your revised manuscript **MUST** contain the following sections before the reference list (for any heading that does not apply to your work, please include a comment to this effect):

- Acknowledgements
- Funding statement

See <https://royalsociety.org/journals/authors/author-guidelines/> for further details.

When uploading your revised files, please make sure that you include the following as we cannot proceed without these:

- 1) A text file of the manuscript (doc, txt, rtf or tex), including the references, tables (including captions) and figure captions. Please remove any tracked changes from the text before submission. PDF files are not an accepted format for the "Main Document".
- 2) A separate electronic file of each figure (tif, eps or print-quality pdf preferred). The format should be produced directly from original creation package, or original software format.
- 3) Electronic Supplementary Material (ESM): all supplementary materials accompanying an accepted article will be treated as in their final form. Note that the Royal Society will not edit or typeset supplementary material and it will be hosted as provided. Please ensure that the supplementary material includes the paper details where possible (authors, article title, journal name). Supplementary files will be published alongside the paper on the journal website and posted on the online figshare repository (<https://figshare.com>). The heading and legend provided for each supplementary file during the submission process will be used to create the figshare page, so please ensure these are accurate and informative so that your files can be found in searches. Files on figshare will be made available approximately one week before the accompanying article so that the supplementary material can be attributed a unique DOI.

Alternatively you may upload a zip folder containing all source files for your manuscript as described above with a PDF as your "Main Document". This should be the full paper as it appears when compiled from the individual files supplied in the zip folder.

Article Funder

Please ensure you fill in the Article Funder question on page 2 to ensure the correct data is collected for FundRef (<http://www.crossref.org/fundref/>).

Media summary

Please ensure you include a short non-technical summary (up to 100 words) of the key findings/importance of your paper. This will be used for to promote your work and marketing purposes (e.g. press releases). The summary should be prepared using the following guidelines:

- *Write simple English: this is intended for the general public. Please explain any essential technical terms in a short and simple manner.
- *Describe (a) the study (b) its key findings and (c) its implications.
- *State why this work is newsworthy, be concise and do not overstate (true 'breakthroughs' are a rarity).
- *Ensure that you include valid contact details for the lead author (institutional address, email address, telephone number).

Cover images

We welcome submissions of images for possible use on the cover of Proceedings A. Images should be square in dimension and please ensure that you obtain all relevant copyright permissions before submitting the image to us. If you would like to submit an image for consideration please send your image to proceedingsa@royalsociety.org

Once again, thank you for submitting your manuscript to Proceedings A and I look forward to receiving your revision. If you have any questions at all, please do not hesitate to get in touch.

Best wishes
 Raminder Shergill
proceedingsa@royalsociety.org
 Proceedings A

Reviewer(s)' Comments to Author:

Referee: 1

Comments to the Author(s)

This paper addresses an important topic in a timely manner. It presents models that can be used to assess the effectiveness of facemasks in managing COVID-19 under various scenarios, including lockdowns. The modelling is appropriate and has been carefully conducted on the basis of reasonable assumptions. The paper is well written and mostly very clear. The paper could well have an influence on government policy. I recommend publication with only very minor changes.

I have inserted comments with suggestions for minor re-wording in the pdf file of the paper. The commented MS is included with this review.

For the non-specialist reader it would help to have a clear definition of terms such as R_e and R_0 when they are first introduced. Then, once the terms are introduced, they should be used consistently, rather than, for example, saying "effective reproductive number".

Referee: 2

Comments to the Author(s)

This is an interesting and timely piece on the added benefits of wearing facemasks as an addition to a lockdown on COVID reproduction rate. The manuscript is well written and clear. It employs two models to make conclusions (a branching model and SIR model).

The authors use facemask effectiveness in influenza outbreaks as an example comparator. Is there any evidence for their use in Tuberculosis, especially in LMIC settings.

When analysing the data, it may be easier to be clarify:

- (1) The model focuses at mask level rather than the droplet dynamic level
- (2) What is the distribution of the mask-status-outcome relationship? Unlikely to be linear, is it discrete distribution?
- (3) Should there be a measure of time of mask worn in a contagious area and probabilistic outcome
- (4) Can there be a measure of efficacy of mask worn? An assumption was made here but can this be updated?
- (5) As this work is focussing on a lockdown+mask, what lessons can we discern from masks without lock down
- (6) There is good modelling evidence of bringing down R from >1 to <1 , however is there any clarity of the use of the mask and its efficacy when R exists below 1 and subsequent results from the model here?

Mask wearing is unlikely to be constant over time – delayed adoption, adoption fatigue etc. Understanding the sensitivity of the models to these human factors would be helpful to determine the applicability of these findings. Similarly, compliance may be inversely correlated with incidence – e.g. at day 300, if there have been very few cases recently, why should I keep wearing a mask?

Modelling

For the SIR model, R_0 is taken to be 4, yet earlier results show R_e is highly dependent on the chosen value of R_0 . It may be useful to include R_0 of 2.2 in the SIR model demonstrate the potential range of benefits arising from mask wearing.

The impact of fomite transmission in the model introduced some counterintuitive findings with non-facemask wearers benefitting more than those wearing a mask. The likelihood of each of the 0, 100 and 300% conditions would be very helpful to understand the most likely scenario encountered, if possible.

Consistency in y-axis ranges in Figure 8 would be helpful for comparison between panels.

Author's Response to Decision Letter for (RSPA-2020-0376.R0)

See Appendix A.

Decision letter (RSPA-2020-0376.R1)

18-May-2020

Dear Dr Stutt

I am pleased to inform you that your manuscript entitled "A modelling framework to assess the likely effectiveness of facemasks in combination with 'lock-down' in managing the COVID-19 pandemic" has been accepted in its final form for publication in Proceedings A.

Our Production Office will be in contact with you in due course. You can expect to receive a proof of your article soon. Please contact the office to let us know if you are likely to be away from e-mail in the near future. If you do not notify us and comments are not received within 5 days of sending the proof, we may publish the paper as it stands.

Under the terms of our licence to publish you may post the author generated postprint (ie. your accepted version not the final typeset version) of your manuscript at any time and this can be made freely available. Postprints can be deposited on a personal or institutional website, or a recognised server/repository. Please note however, that the reporting of postprints is subject to a media embargo, and that the status the manuscript should be made clear. Upon publication of the definitive version on the publisher's site, full details and a link should be added.

You can cite the article in advance of publication using its DOI. The DOI will take the form: 10.1098/rspa.XXXX.YYYY, where XXXX and YYYY are the last 8 digits of your manuscript number (eg. if your manuscript number is RSPA-2017-1234 the DOI would be 10.1098/rspa.2017.1234).

For tips on promoting your accepted paper see our blog post:
<https://blogs.royalsociety.org/publishing/promoting-your-latest-paper-and-tracking-your-results/>

On behalf of the Editor of Proceedings A, we look forward to your continued contributions to the Journal.

Sincerely,
Raminder Shergill
proceedingsa@royalsociety.org

Appendix A

A modelling framework to assess the likely effectiveness of facemasks in combination with 'lock-down' in managing the COVID-19 pandemic

Stutt, Retkute, Bradley, Gilligan, Colvin

Response by authors to peer review comments

Reviewer #1

This paper addresses an important topic in a timely manner. It presents models that can be used to assess the effectiveness of facemasks in managing COVID-19 under various scenarios, including lockdowns. The modelling is appropriate and has been carefully conducted on the basis of reasonable assumptions. The paper is well written and mostly very clear. The paper could well have an influence on government policy. I recommend publication with only very minor changes.

We thank the reviewer for the careful reading of the paper.

I have inserted comments with suggestions for minor re-wording in the pdf file of the paper. The commented MS is included with this review.

We have addressed/incorporated all of the recommended changes and edits and are grateful to the reviewer for reading the paper through in such detail.

For the non-specialist reader it would help to have a clear definition of terms such as R_e and R_0 when they are first introduced. Then, once the terms are introduced, they should be used consistently, rather than, for example, saying "effective reproductive number".

We have now introduced clear definitions as requested by the reviewer.

Reviewer #2

This is an interesting and timely piece on the added benefits of wearing facemasks as an addition to a lockdown on COVID reproduction rate. The manuscript is well written and clear. It employs two models to make conclusions (a branching model and SIR model).

We are grateful to the reviewer for detailed comments to help clarify the manuscript and to highlight possibilities for further work.

The authors use facemask effectiveness in influenza outbreaks as an example comparator. Is there any evidence for their use in Tuberculosis, especially in LMIC settings.

We thank the reviewer for his/her question about evidence for facemask use in Tuberculosis, especially in LMIC settings. In preparing our paper, we reviewed the evidence for TB, MERS and SARS and there are indeed recommendations for facemask use by healthcare workers to provide protection against all three diseases. The transmission characteristics of these three causative pathogens differ from each other and from SARS-CoV-2. MERS was strikingly different, for example, in that there wasn't sustained person-to-person human transmission reported (with or without facemask use). We prefer, therefore, not to introduce these other pathogens into our paper, where a key feature of SARS-CoV-2 is the infectiousness of non-symptomatic individuals.

When analysing the data, it may be easier to be clarify:

(1)The model focuses at mask level rather than the droplet dynamic level

Action: we have introduced a phrase in the introduction to make the distinction and focus of the paper clear.

(2)What is the distribution of the mask-status-outcome relationship? Unlikely to be linear, is it discrete distribution?

The system is indeed non-linear and we show trade-offs to mask and non-mask wearers in Fig. 8. It is not clear to us what the reviewer was looking for querying about a discrete distribution. We believe, however, that we cover the main point in Fig. 8 and elsewhere in Figs 5-6, which show the non-linear impact on infection levels from wearing masks. No further action taken to revise the manuscript.

(3)Should there be a measure of time of mask worn in a contagious area and probabalistic outcome

In the current paper, we consider the efficacy of mask wearing and the proposition of the population wearing masks. We indicate in the text that the SIR model could be readily adapted for metapopulations, for example as referred to by the reviewer to consider highly contagious areas (e.g. workplace/shops) with lower contagious areas (external environment in cities). We could show differences here but, in the absence of suitable parametres, we consider this best referred to as one amongst several possibilities for further work, as more data become available. **Action:** we pick this up in the discussion.

Reviewer # 2 indicated a number of additional scenarios that could be addressed by the modelling framework introduced in the current paper. These include spatial partitioning of regions based on risk of contact, behavioural shifts based on individual perception of risk and inconsistent use of masks, all of which are potentially important. We suggest, however, that these should form the basis of future work, with priorities for investigation set jointly by policy demands and the availability of additional data for parameterisation.

(4) can there be a measure of efficacy of mask worn? An assumption was made here but can this be updated?

We do include a measure of mask efficacy as the reviewer indicates in our simulations (Figs 3,4 and Figs 5-8) and the efficacy could be updated, for example to allow for a changing availability of masks of different design. **Action:** we have added a note to this effect in the discussion.

See also response to reviewer's comments under item 6 below.

(5)As this work is focussing on a lockdown+mask, what lessons can we discern from masks without lock down

We agree with the reviewer that this is important and we do indeed show the effects of masks in the absence of lockdown for both models (Figs 3 and 4 for Model 1 and Figs 7 & 8 for Model 2). We state in the original ms that "For completeness, we analysed the effects of facemask use in the absence of 'lock-down' or other mitigation procedures (Figure 7)" We also discuss the issue further on p16 of the original and revised ms, so have made no further changes.

(6) There is good modelling evidence of bringing down R from >1 to <1 , however is there any clarity of the use of the mask and its efficacy when R exists below 1 and subsequent results from the model here?

This is an interesting observation and it has led us to examine and comment further on the results in Fig. 6. We compare Figs 6b and 6d for both of which the $R_e < 1$ and the comparison shows how introducing mask wearing earlier (Fig. 6b) brings the epidemic down faster than later introduction (Fig. 6d). **Action:** we have added a comment to this effect in the Results section. We also note that when the numbers of infected becomes small, contact tracing is likely to be pursued rather than mask wearing.

Mask wearing is unlikely to be constant over time – delayed adoption, adoption fatigue etc. Understanding the sensitivity of the models to these human factors would be helpful to determine the applicability of these findings. Similarly, compliance may be inversely correlated with incidence – e.g. at day 300, if there have been very few cases recently, why should I keep wearing a mask?

We agree with the reviewer that these are important practical considerations and we now introduce these pointers for further work in the Discussion. Our motivation and the core of this research is to introduce and demonstrate key results from the modelling framework, which we have done here. While there are a large number of other simulations that could be done, we prefer not to expand the paper unduly with additional simulations.

Modelling

For the SIR model, R_0 is taken to be 4, yet earlier results show R_e is highly dependent on the chosen value of R_0 . It may be useful to include R_0 of 2.2 in the SIR model demonstrate the potential range of benefits arising from mask wearing.

We do consider $R_0 = 2.2$ and 4.0 for Model 1. We also carried the equivalent simulations for Model 2 (Figs 5-7) for $R_0 = 2.5$ as well as $R_0 = 4$. Given that we show an effect for Model 2 at $R_0 = 4$, the effects will be greatly enhanced for $R_0 = 2.2$ (revealed in our simulations). **Action:** we now make a comment to this effect in the Discussion.

The impact of fomite transmission in the model introduced some counterintuitive findings with non-facemask wearers benefitting more than those wearing a mask. The likelihood of each of the 0, 100 and 300% conditions would be very helpful to understand the most likely scenario encountered, if possible.

This is a good point but, unfortunately, in the absence of evidence it is impossible to identify the most likely scenario. **Action:** we have added a comment in the Discussion to emphasise the need for experimental evidence on the likely values.

Consistency in y-axis ranges in Figure 8 would be helpful for comparison between panels.

We did originally set all panels to the same scale, but this occluded the comparisons between mask-wearers, non-wearers and the total population within each panel. **Action:** we prefer to keep it as it is but we have added a comment in the figure caption to note that the vertical axes scale over different ranges, to show clearly the distinctions between mask-wearers, non-wearers and the total population, for each scenario.

13th May 2020